# The Lung Microbiome in COPD and Lung Cancer: Exploring the Potential of Metal-Based Drugs

**DOI:** 10.3390/ijms241512296

**Published:** 2023-08-01

**Authors:** Megan O’Shaughnessy, Orla Sheils, Anne-Marie Baird

**Affiliations:** 1School of Medicine, Trinity Translational Medicine Institute, Trinity College Dublin, D08 W9RT Dublin, Ireland; 2Department of Histopathology and Morbid Anatomy, Trinity Translational Medicine Institute, St. James’s Hospital, D08 RX0X Dublin, Ireland

**Keywords:** lung microbiome, chronic obstructive pulmonary disease, lung cancer, metal-based drugs, phenanthroline, metal-phen complexes

## Abstract

Chronic obstructive pulmonary disease (COPD) and lung cancer 17 are two of the most prevalent and debilitating respiratory diseases worldwide, both associated with high morbidity and mortality rates. As major global health concerns, they impose a substantial burden on patients, healthcare systems, and society at large. Despite their distinct aetiologies, lung cancer and COPD share common risk factors, clinical features, and pathological pathways, which have spurred increasing research interest in their co-occurrence. One area of particular interest is the role of the lung microbiome in the development and progression of these diseases, including the transition from COPD to lung cancer. Exploring novel therapeutic strategies, such as metal-based drugs, offers a potential avenue for targeting the microbiome in these diseases to improve patient outcomes. This review aims to provide an overview of the current understanding of the lung microbiome, with a particular emphasis on COPD and lung cancer, and to discuss the potential of metal-based drugs as a therapeutic strategy for these conditions, specifically concerning targeting the microbiome.

## 1. Introduction

The human microbiome has gained significant attention due to its influential role in health and disease [1,2]. While there has been extensive research on gut microbiota, the microbiome of the human respiratory tract has received less attention, mainly because the lungs were historically considered sterile [3]. The advent of modern culture-independent methods and next-generation sequencing (NGS) techniques has heralded a paradigm shift in our comprehension of the microbial landscape within the lung [4,5]. This realisation has sparked a burgeoning field of research, unveiling fascinating insights into the complex interplay between the lung microbiome, the immune system, and host physiology [6,7]. The lung microbiome exhibits remarkable plasticity, dynamically shifting in response to a myriad of factors, including age, environmental exposure, lifestyle choices, medications, and genetic predisposition [4,7,8,9]. Evidence from human studies has shown that alterations in the lung microbiome have been implicated in several pathological pulmonary conditions, including chronic obstructive pulmonary disease (COPD) and lung cancer [10,11,12].

In 2019, 212.3 million cases of COPD were reported globally, with COPD accounting for 3.3 million deaths [13]. The World Health Organization (WHO) predicted that COPD would be the third leading cause of death by 2030. However, it has already reached that landmark and is now expected to be the most lethal respiratory disease within the same time frame [14]. Lung cancer is the second most commonly diagnosed cancer and is responsible for more cancer-related deaths globally than any other cancer type [15]. The overall five-year survival rate is approximately 20%, which is partly due to the majority of people presenting with advanced disease. There is a high incidence of lung cancer among people with COPD, and this association transcends the established connection that both conditions have with a smoking history. For instance, several studies have reported a heightened risk of lung cancer in people with COPD, independent of age or tobacco use [16,17,18,19,20,21,22,23]. The data suggest that COPD in those who smoke have a two to five-fold higher risk of developing lung cancer compared to non-smokers, with the overall survival rate in those with COPD and lung cancer significantly lower than those with lung cancer without COPD [17,24,25]. Park et al. [23] undertook a large national cohort study in South Korea that investigated the incidence rate of lung cancer in never-smokers with COPD. In comparison to never-smokers without COPD, the hazard ratios for lung cancer in never-smokers with COPD, ever-smokers without COPD, and ever-smokers with COPD were 2.67, 1.97, and 6.19, respectively, indicating that COPD is a strong independent risk factor of lung cancer, irrespective of smoking status [23]. Reported mechanisms linking COPD and lung cancer include genetic factors, epigenetic changes, non-coding RNAs, oxidative stress, immune environment, and sex differences [17,26,27,28,29]. However, emerging evidence also suggests that dysregulation in the lung microbiome may play a key role in the pathogenesis of COPD and the transition to lung cancer [7,12,30]. In relation to the microbiome, commonly reported mechanisms are the chronic inflammatory environment in COPD, fuelled by an imbalanced lung microbiota, which can create a pro-oncogenic environment promoting lung cancer through oxidative stress, genetic mutation, and DNA damage [26]. Dysregulated lung microbiota can also modulate the immune response, with certain bacteria inducing pro-inflammatory cytokines, potentially fostering lung cancer development [31]. The precise pathogenic mechanisms through which the microbiome mediates the progression of lung cancer remain largely elusive, although they are considered multifaceted and involve bacterial toxins such as lipopolysaccharide (LPS) and the release of inflammatory cytokines by immune cells [32]. For instance, the LPS of *H. pylori* can trigger the production of pro-inflammatory agents, including tumour necrosis factor (TNF), interleukin (IL)-1, and IL-6 [33]. These inflammation-promoting mediators are implicated in the progression of chronic lung diseases such as COPD as a precursor to lung cancer [27]. Understanding the role of the lung microbiome in the pathogenesis and progression of COPD to lung cancer may also open new avenues for therapeutic interventions, such as metal-based drugs.

The use of metals and metal-based drugs in medicine has a longstanding history, with some of the earliest examples dating back to ancient civilisations [34,35]. The potential of metal-based drugs to exhibit unique biochemical properties, largely due to the diverse redox states and coordination geometries of metal ions, allows for a wide range of potential applications, including antimicrobial agents [36,37,38,39,40], cancer therapeutics [41,42,43], and modulators of the microbiome [44,45,46]. Moreover, their ability to engage in multiple simultaneous interactions with biomolecules increases their therapeutic effectiveness and versatility [36,47]. Recent research has shown that metal-based drugs can manipulate microbial populations, both in vitro and in vivo, by selectively inhibiting growth or suppressing virulence in a multi-modal fashion [37,38,48]. For instance, gold-based drugs have demonstrated antimicrobial effects and can modulate gut microbiota [49,50,51]. In the context of the lung, the use of metal-based drugs as microbiome modulators is still in its infancy, but could offer a new avenue in treating respiratory diseases where chronic infections and dysregulation of the lung microbiome play a critical role. 

## 2. The Healthy Lung Microbiome

The concept of the lung microbiome has been radically altered with the understanding that the human lung hosts a complex ecosystem comprised of bacteria, fungi, and viruses, which actively participate in maintaining respiratory health by contributing to immune modulation, pathogen displacement, and metabolic contributions [52,53]. The composition of the respiratory microbiome is transient and determined by continuous microbial immigration (through microaspiration, inhalation, and direct mucosal spread), elimination (by the immune system and mucociliary clearance), and replication [3]. The notion of a healthy lung microbiome refers to a state in which a multitude of beneficial microorganisms coexist in harmony, promoting an immune environment that is neither too reactive nor too lax, providing robustness against invading pathogens, and supporting the crucial function of the lungs [5,54]. Under homeostatic conditions, the predominant bacterial phyla in the human lung are Bacteroidota (45–50%, with the genus *Prevotella*), Bacillota (30–35%, including the genera *Streptococcus* and *Veillonella*), Pseudomonadota (10–15%, with the genera *Haemophilus* and *Neisseria*), Actinomycetota (5%, represented by the genus *Corynebacterium*), and Fusobacteriota (5%, with the genus *Fusobacterium*) [7,55]. The diversity of the microbiome, which is a significant factor in maintaining respiratory health, is reflected by its richness (total number of different microbial species) and its evenness (relative abundance or proportion of different species) [3,56]. In a diseased state, alterations in the structure and local microenvironment of the lung, such as changes in mucosal pH, oxygen gradients, nutrient availability, inflammation, and host defence, foster proliferation of pathogenic microbes and consequently lead to shifts in the microbiome composition [5,57,58]. These disruptions, known as dysbiosis, can have detrimental effects on lung health. Dysbiosis can result from various factors, including environmental influences (exposure to harmful pollutants or smoking), antibiotic use, respiratory infections, lifestyle, or genetics, and has been associated with several pulmonary diseases, including COPD and lung cancer [7,58]. Understanding the factors that contribute to dysbiosis and developing strategies to restore a healthy lung microbiome through targeted approaches could offer novel therapeutic avenues for these conditions. 

## 3. The Lung Microbiome in Lung Disease

Numerous studies have distinguished the lung microbiome in human health and disease, in which a shift of the microbiome is associated with diseases and key clinical parameters, such as severity, exacerbation, phenotype, endotype, inflammation, and mortality [57,59,60,61]. Dysbiosis, characterised by a decrease in microbial diversity and a shift in community composition, is observed in various respiratory disorders, including asthma, cystic fibrosis, idiopathic pulmonary fibrosis (IPF), tuberculosis, COPD, and lung cancer [62,63,64,65,66,67]. Diversity is a measure of the evenness and richness of a microbial community, which can be measured within a biological sample (α-diversity) or between samples (β-diversity) [68,69]. Lower bacterial diversity has been linked to disease progression, although it is unknown whether microbial dysbiosis is a cause or effect of the disease [2,7,70]. Dysbiosis may have a causative role in lung diseases by upregulating inflammatory signals (such as nuclear factor kappa-light-chain-enhancer of activated B cells (NF-κB), Ras, IL-17, and phosphoinositide 3-kinase (PI3K)) [67,71,72,73,74] or by suppressing the production of TNF and interferon-gamma (IFNγ) in response to pathogen presence in the lower respiratory tract [58,75]. The following sections will focus specifically on the lung microbiome in COPD and lung cancer. 

### 3.1. Chronic Obstructive Pulmonary Disease (COPD)

COPD is a chronic inflammatory lung condition characterised by persistent respiratory symptoms and progressive airflow restriction. Predominantly triggered by long-term exposure to harmful pollutants, such as cigarette smoke and environmental toxins, it clinically presents with dyspnoea, chronic cough, sputum production, and wheezing [76,77]. The disease comprises two primary phenotypes: chronic bronchitis, hallmarked by a chronic productive cough, and emphysema, typified by alveolar wall destruction over time. COPD, as a progressive and potentially fatal condition, can significantly deteriorate quality of life due to recurrent exacerbations and a decline in pulmonary function [77,78,79]. Although there is no definitive cure at present, the management of symptoms and decelerating disease progression are pivotal. Recent research has begun to shed light on the role of the lung microbiome in the pathogenesis and progression of COPD, offering a new perspective on this debilitating disease [80,81,82]. Gram-negative pathogenic bacteria tied to COPD (*Haemophilus* spp., *Moraxella*, *Pseudomonas*) possess a notably higher potential to stimulate an immune response compared to Gram-negative commensal bacteria (*Prevotella* spp.) [83]. Poor oral hygiene has been identified as a risk factor for inflammatory lung conditions through microaspiration of oral commensals, such as *Veillonella* and *Prevotella*, which have been associated with increased T_H_17 lymphocytes within the lung [71,84]. Several studies have reported the composition of the airway microbiome in COPD and found a shift in microbiome diversity with a decrease in Bacillota (Firmicutes) and Bacteroidota and an increase in Pseudomonadota, particularly the genus *Haemophilus*, which positively correlated with IL-8 present in sputum [8,80,85,86,87,88]. This shift in the microbiome composition, particularly the elevation in Pseudomonadota, has been associated with greater emphysema, and increased immune cell infiltration leading to chronic inflammation, airway remodelling, and exacerbations [88,89,90,91,92]. Exacerbations, defined as acute worsening of respiratory symptoms, significantly contribute to the morbidity and mortality associated with COPD. These episodes are often triggered by bacterial or viral infections and treated with antibiotics and corticosteroids, critical elements in the standard therapeutic approach [93]. Antibiotic-mediated perturbation of the gut microbiome has been widely reported to be associated with numerous infectious and autoimmune diseases of the gastrointestinal tract [94,95,96,97]. Although less extensively studied, antibiotic use has been reported to cause alterations in the lung microbiome, which can negatively impact the ecological balance of microbial communities within the lung and potentially escalate disease progression and exacerbation severity [56,98]. Moreover, dysbiosis may potentiate bacterial resistance, creating challenges for future antimicrobial treatment [99,100]. Thus, while antibiotics and corticosteroids are essential for managing COPD exacerbations, their impact on the lung microbiome warrants careful consideration within therapeutic strategies [101]. 

The lung microbiome in COPD has been found to be significantly associated with bacterial biomass, lymphocyte proportion, T_H_17 immune response, exacerbation frequency, and resistance to antimicrobial therapy [102]. Initial investigations into the microbiome of patients with stable COPD have demonstrated a significant correlation between the presence of pathogenic bacteria, such as *Haemophilus influenzae*, *Streptococcus pneumoniae*, *Moraxella catarrhalis*, *Staphylococcus aureus*, *Pseudomonas aeruginosa*, and Enterobacterales. As COPD progresses, chronic inflammation impairs the innate immune response within the lung, which in turn creates a favourable environment for an increase in bacterial burden. Numerous studies have shown that in moderate to severe COPD (Global Initiative for COPD (GOLD) 2–4), there is an enrichment of Gammaproteobacteria (*Haemophilus* and *Moraxella* spp.) in bronchiole lavage and lung tissue samples [80,103,104]. Erb-Downward et al. [80] found that the microbiomes of patients with moderate or severe COPD had lower bacterial diversity scores than healthy smokers and non-smokers. They identified a core COPD lung microbiome that included *Pseudomonas*, *Streptococcus*, *Prevotella*, *Fusobacterium*, *Haemophilus*, *Veillonella*, and *Porphyromonas* [80]. Interestingly, in patients with very advanced COPD, there were significant differences in the microbiomes at adjacent lung sites, suggesting heterogeneity within the individual lung microbiome. In a similar vein, Sze et al. [105] evaluated the microbiomes in lung tissues taken from patients with severe COPD (GOLD 4) at the time of lung transplantation and found an increase in bacterial diversity in patients with severe COPD compared to non-smokers, smokers, and patients with CF, with a notable increase in the phylum Bacillota (Firmicutes), specifically *Lactobacillus* [105]. These findings highlight the dynamic nature of the lung microbiome in COPD patients and the differences across sample types. In a longitudinal observational study analysing the microbiome of sputum of clinically stable COPD patients, reduced microbiome diversity was observed associated with Pseudomonadota (predominantly *Haemophilus*) dominance, which was associated with neutrophil-associated protein profiles and an increased risk of mortality [89]. In a recent cohort study, longitudinal sputum samples were taken from COPD patients during acute exacerbation (AECOPD), which found significant positive correlations between the abundance of *Pseudomonas* and TNF, the abundance of *Klebsiella,* and the percentage of eosinophils. Furthermore, the study identified four clusters of COPD based on the respiratory microbiome, with the AECOPD-related cluster characterised by the enrichment of *Pseudomonas* and *Haemophilus* and a high level of TNF [88]. Therefore, these patients might benefit from targeted antibacterial agents, which may aid in alleviating inflammation. In contrast, patients with a diverse microbiome profile, including *Veillonella* and *Prevotella*, exhibited a more dynamic microbiome over time and showed elevations of IL-17A within sputum and serum [106]. It was observed that these patients had greater microbiome shifts during exacerbations and, therefore, would profit from an anti-inflammatory therapeutic strategy. This growing body of evidence underscores the intricate relationship between the lung microbiome and COPD, shedding light on the potential of microbiota-targeted interventions to improve the long-term prognosis of COPD. However, a comprehensive understanding of these relationships warrants further in-depth investigations, including longitudinal studies to track microbiome changes over time and interventional studies to test the causality of observed associations.

### 3.2. Lung Cancer

Lung cancer, a heterogeneous group of malignancies arising in the lung parenchyma or bronchi, is the leading cause of cancer-related deaths globally [107]. The disease is classified into non-small cell lung cancer (NSCLC), which constitutes about 85% of cases, and small cell lung cancer (SCLC), which represents the remaining 15%. The three main subtypes of NSCLC are adenocarcinoma (40%), squamous cell carcinoma (25–30%), and large cell carcinoma (5–10%). Lung cancer is typically asymptomatic in the early stages, with clinical manifestations such as persistent cough, chest pain, and haemoptysis emerging as the disease progresses [108,109,110]. The predominant contributors to lung cancer include exposure to tobacco smoke, environmental toxins, carcinogens, persistent airway inflammation instigated by pathogenic infections, as well as fibrosis and scarring resulting from co-existing lung diseases [29,111]. Despite advancements in therapeutic interventions, lung cancer continues to be a primary cause of death related to cancer worldwide due to its late-stage diagnosis and high recurrence rates, even in those with early-stage disease. Comprehensive strategies for early detection and treatment are crucial to reducing the global burden of the disease. 

The intricate relationship between the lung microbiome and lung cancer has begun to be elucidated. Research has shown associations between specific bacterial phyla, family, genera, and species and the progression of lung cancer. Among these are Actinomycetota, Bacteroidota, Pseudomonadota, Bacillota (Firmicutes) [112], Capnocytophaga, Neisseria, and Selenomonas [113], *Thermus*, *Legionella*, *Megasphaera*, *Veillonella* [114], *Cyanobacteria* [115], *Acidovorax temporans* [33] and *Helicobacter pylori* [116]. Dysbiosis of the microbiome is mainly manifested by the decrease in symbiotic bacteria and the increase in pathogenic bacteria, and then inducement of carcinogenesis at multiple levels, including metabolism alteration, inflammation, and altered immune response [117,118]. Several studies have pointed towards an increased abundance of certain genera, such as *Veillonella*, *Streptococcus*, and *Prevotella,* in lung cancer patients compared to healthy individuals, which have also been observed in patients with COPD [119,120]. For example, a study by Yu et al. [121] reported that the microbiota of lung tumour tissue showed significantly lower α-diversity compared to non-malignant lung tissue samples, with higher levels of *Veillonella*. Moreover, it was reported that bacterial composition correlated with cancer stage, with a higher abundance of *Thurmus* in advanced-stage (IIIB, IV) patients, and *Legionella* in patients with metastases [121]. Studies have also indicated an enrichment of *Veillonella* and *Megasphaera* in the bronchoalveolar lavage fluid (BALF) of patients diagnosed with NSCLC stages II-IV (72% adenocarcinoma and 28% squamous cell carcinoma), suggesting these genera could serve as biomarkers to predict disease progression [113]. Complementary findings by Huang et al. [122] support this observation in the bronchial washing fluid of patients with lung adenocarcinoma compared to those with squamous cell lung carcinoma. Gomes et al. [114] reported that lung cancer microbiota was enriched in Pseudomonadota and more diverse in squamous cell carcinoma than adenocarcinoma, particularly in males and heavier smokers, suggesting a potential link between these bacteria and the presence of other risk factors. Najafi et al. [123] found that the relative abundance of several bacterial taxa, including Actinomycetota phylum, *Corynebacteriaceae,* and *Halomonadaceae* families, and *Corynebacterium*, *Lachnoanaerobaculum*, and *Halomonas* genera, is significantly decreased in lung tumour tissues of lung cancer patients in comparison with matched tumour-adjacent normal tissues. The microbiota within the airway plays a distinct role in the onset and progression of lung cancer. Research has indicated that mice, either germ-free or treated with antibiotics, show considerable resistance against lung cancer development, even with Kras mutation and p53 loss [124]. Other studies have identified differences in the gut microbiota between patients with lung cancer and healthy controls, with an increased abundance of *Enterococcus* and decreased levels of the phylum Actinomycetota and genus *Bifidobacterium* [125]. The “gut-lung axis” is an emerging concept linking the state of the gut microbiota to respiratory health outcomes [126,127]. Simultaneously, it was observed that depleting the microbiota or inhibiting γδ T cells or their downstream effector molecules all effectively suppressed the growth of lung adenocarcinoma in a genetically engineered mouse model driven by an activating point mutation of Kras and loss of p53 [128]. The role of the resident microbiome in the progression of lung cancer was further examined in a nested case control study encompassing 4336 lung cancer subjects and 10,000 matched controls aged 40–84 years [129]. The study sought to establish any correlation between antibiotic usage and the risk of lung cancer. Remarkably, subjects who received 10 or more antibiotic courses presented a relative lung cancer risk of 2.52 (95% CI, 2.25–2.83) compared with controls who had not received antibiotics. The elevated relative risk could potentially be attributed to the inflammatory conditions induced by frequent infections and consequent alterations in the lung microbiomes among those administered antibiotics [129].

The proposed theory is that dysbiosis or microbial imbalance might propel carcinogenesis through three channels: (i) disruption of immune equilibrium, (ii) chronic inflammation instigation, and (iii) activation of cancer-causing pathways [130,131,132,133,134]. Firstly, dysbiosis has the potential to disrupt the lung immune system’s fundamental stimulation, and depletion of microbial diversity impairs the initial activation of antigen-presenting cells, thereby inhibiting their response to tumour antigens [2]. Conversely, bacterial overgrowth can lead to an overstimulation of the immune system and unchecked proliferation of IL-17-producing CD4^+^ helper T (T_H_17) cells, mediators in lung tumorigenesis [135]. Secondly, dysbiosis incites chronic inflammation via the release of DNA-damaging metabolites and genotoxins from commensal organisms. Inflammatory cells activated by dysbiosis can also release reactive oxygen (ROS) and nitrogen (RNS) species, promoting carcinogenesis and angiogenesis [136,137]. Lastly, several studies have highlighted that some species within the microbiota can directly stimulate cancer-causing pathways. For instance, Apopa et al. [115] demonstrated an increase in PARP1 in NSCLC tissues in the presence of the *cyanobacteria* toxin microcystin. Likewise, research by Tsay et al. [67] connected *Streptococcus* and *Veillonella* to the stimulation of the PI3K and extracellular signal-regulated protein kinase (ERK) pathways involved in the disease. Ochoa et al. [138] found that exposure of the airway to smoke particulates and nontypeable *H. influenzae* (NTHi) promoted lung cancer cell proliferation by release of IL-6 and TNF, which further activated the STAT3 and NF-κB pathways in the airway epithelium. Interestingly, it was demonstrated that IL-6 blockade significantly inhibited lung cancer promotion, tumour cell-intrinsic STAT3 activation, tumour cell proliferation, and angiogenesis markers [139]. As previously mentioned, T_H_17 cell-mediated inflammation has been identified as playing a critical role in lung tumorigenesis [140]. Jungnickel et al. [73] indicated that the epithelial cytokine IL-17C mediates the tumour-promoting effect of bacteria, such as NTHi, through neutrophilic inflammation. There has been growing awareness of the importance of NTHi in the pathophysiology of COPD, and COPD-like airway inflammation induced by NTHi provides a tumour microenvironment that favours cancer promotion and progression [141,142,143]. Thus, NTHi may act as a bridge between COPD and lung cancer. Despite these encouraging findings associating specific microbes with carcinogenesis, distinguishing between microbes genuinely inducing cancer-causing pathways and those opportunistically colonising the tumour microenvironment remains a challenge. However, given the impact of dysregulated lung microbiomes on diseases such as COPD and lung cancer, the exploration of innovative therapeutics is warranted. Metal-based drugs offer a promising avenue with their potential to modulate the microbiome, alleviate inflammation, and directly target malignant cells.

## 4. Novel Therapeutic Strategies: Metal-Based Drugs

Metal-based remedies have played a crucial role in medicine throughout history. Ancient Egyptians discovered the therapeutic potential of gold (Au) salts, and alchemists mixed powdered Au into water to ease the aching of limbs, which is one of the primal references to arthritis [144]. Chinese medicine has long established the antiseptic competence of arsenic (As) [145], and the use of silver (Ag) in wound management can be traced back to the 17th century, during which silver nitrate (AgNO_3_) was administered to treat ulcers [146]. Paul Ehrlich and his co-workers developed the first successful metalloid complex in the 1900s, an As-based therapeutic agent named salvarsan. This effectively cleared a syphilis infection for which no prior treatment was available [147,148]. Nevertheless, research into metal-based coordination complexes was not stimulated until the fortuitous discovery of a platinum (Pt) complex known as cisplatin by Barnett Rosenberg [149]. Since then, cisplatin has been extensively studied as a chemotherapeutic drug and is broadly administered to various cancers, including ovarian, lung, head and neck, testicular, and bladder [150,151]. It is particularly effective in testicular germ cell tumours, leading to complete remission in approximately 70–80% of treated patients [152], and combination treatment with radiotherapy is more successful than radiotherapy alone in cohorts of NSCLC [153]. Cisplatin is structurally a coordination compound with square planar geometry and exerts its antitumour activity via intra-cellular hydrolysis and subsequent covalent binding to DNA-forming adducts, triggering apoptosis [154]. Despite its triumph, cisplatin has several disadvantages, including toxicity and drug resistance [155,156,157]. Since the discovery of cisplatin, a range of metal-containing drugs have been approved to treat various conditions, including cancer, rheumatoid arthritis, anaemia, iron overload, bipolar disorder, and gastrointestinal disorders [42] as presented in Table 1.

The realm of metal-based drugs also holds substantial promise in the treatment of lung cancer, offering novel mechanisms of action that may help overcome the limitations of traditional therapies [43,158,159,160]. The unique properties of metals, bring forth new avenues for attacking malignant cells while mitigating side effects associated with conventional cancer treatment [161]. Despite significant advances in personalised medicine and the development of therapies specifically targeting driver mutations, Pt-based chemotherapy remains a viable option in the management of NSCLC [162]. Cisplatin, the pioneer in this class, has been extensively used as a first-line therapy for NSCLC, and carboplatin, a less nephrotoxic and neurotoxic derivative of cisplatin, is frequently administered in conjunction with other agents as part of combined chemotherapeutic protocols for NSCLC [163,164]. A significant clinical challenge in the treatment of NSCLC patients with Pt agents, in particular cisplatin, is intrinsic and acquired resistance, which necessitates alternative metal-based drugs [165]. Ruthenium (Ru) complexes, such as KP1019 and NAMI-A, have shown potential in clinical studies in patients with NSCLC [166,167]. Ru(III)-based complexes can mimic iron, allowing them to be taken up by cells where Ru(III) can be reduced to Ru(II) in the hypoxic environments typical of solid tumours, which then irreversibly bind to DNA and proteins, inducing cytotoxic effects [168]. Another emerging class of metal-based drugs involves Au(I) complexes, such as auranofin, which have been shown to exhibit anticancer activity due to their ability to inhibit the selenoenzyme thioredoxin reductase (TrxR) [169]. A phase II clinical trial which assessed auranofin with sirolimus in patients with advanced or recurrent NSCLC found the combination arrested and regressed tumours [170]. Copper (Cu) complexes, such as Casiopeinas, have demonstrated anticancer properties through mechanisms such as DNA damage, ROS generation, and topoisomerase II inhibition [171]. Tetrathiomolybdate, a copper chelator, was assessed in a phase I clinical trial for NSCLC; however, it was found to be more effective in breast cancer targeting the copper-dependent components of the tumour microenvironment [172]. Additionally, there is growing interest in gallium (Ga) compounds due to their ability to disrupt the iron-dependent processes critical for cancer cell growth. For instance, gallium nitrate has shown potential in combating NSCLC [173]. While the application of metal-based drugs in the treatment of lung disease exhibits promise, there are several limiting factors that need to be addressed [174]. One major limitation lies in the potential adverse effects and systemic toxicity associated with these therapeutics. Such compounds often display indiscriminate toxicity towards both cancerous and healthy cells, leading to a host of side effects, including toxicity, and myelosuppression, which can severely compromise patient health and limit the dosage that can be safely administered [175,176]. Furthermore, the issue of drug resistance often surfaces with prolonged use of these agents, mirroring the challenges faced with more traditional chemotherapeutic agents [177]. The pharmacokinetics of metal-based drugs can also present limitations; poor bioavailability and non-specific biodistribution can affect drug efficacy and increase the potential for systemic toxicity [178,179]. The future of metal-based drugs in lung cancer treatment will depend on concerted efforts in research, clinical trials, and the translation of findings to clinical practice.

## 5. Metal Drugs as Microbiome Modulators

The current antimicrobial clinical pipeline is inadequate to treat mounting infections caused by multidrug-resistant pathogens [180]. Of the 12 new antibacterial agents approved for clinical use since 2017, the WHO reported that only one compound, cefiderocol, meets their innovation criteria (absence of known cross-resistance, new target, a new mode of action, or new class) that also has activity against all three critical priority pathogens [181]. Cefiderocol is a siderophore-conjugated cephalosporin that promotes the formation of chelated complexes with ferric iron and facilitates siderophore-like transport across the outer membrane of Gram-negative bacteria using iron transport systems accumulating in the periplasmic space [182,183]. This has highlighted the essential need to investigate ‘non-traditional’ approaches to antibacterial therapy that explore different avenues compared to ‘traditional’ organic molecules that target pathogens through already established targets [184,185]. These agents can prevent or treat bacteria through several modes of action, including directly or indirectly inhibiting growth, dampening virulence, truncating, or removing biofilm, alleviating resistance, restoring the natural microbiome, or boosting the immune system to clear or manage infections [184,186,187]. 

Metal-bearing drugs can adopt a range of coordination geometries and redox states, allowing for more significant chemical variations when compared with purely organic antibiotics, with different and potentially multi-modal mechanisms of action [40,179,188]. For instance, a recent study by the Community for Open Antimicrobial Drug Discovery (CO-ADD), a global free open-access screening initiative, discussed metal complexes’ enhanced activity profile [189]. The group evaluated 906 individual metal compounds within their database, from d-block elements, against critical ESKAPE bacteria and fungi, and found an impressive success rate of the metal compounds (9.9%) in comparison to solely organic molecules (0.87%). From this panel of metal complexes, 88 demonstrated activity (minimum inhibitory concentration (MIC) ≤ 16 µg/mL or 10 µM) against one of their tested strains (58 against fungi and 30 against bacteria) while also being tolerated by mammalian cells (CC_50_ > 32 µg/mL or >20 µM against human embryonic kidney cell line) and not demonstrating haemolytic activity (HC_10_ > 32 µg/mL or >20 µM). Only 14 of these metal complexes showed activity (MIC ≤ 32 µg/mL) against Gram-negative bacteria, including pathogenic bacteria tied to COPD and lung cancer [83,114]. Overall, the group emphasised the potential therapeutic capabilities of metal compounds due to the extent of possible modes of action, with broader coverage of three-dimensional chemical space than their organic counterparts [189]. 

The diverse antimicrobial mechanisms of metal-based drugs offer a promising avenue of exploration for the treatment of respiratory diseases, where dysbiosis plays a significant role, including COPD and lung cancer. The subsequent sections will focus on notable discoveries of metal-based compounds that have been studied as antimicrobial agents and exhibit the characteristics of microbiome modulators. However, it is in no way exhaustive. The coordination of metals to organic ligands such as 1,10-phenanthroline (phen) will also be discussed, as metal-phen complexes are emerging as tangible alternatives to the traditional antibiotic, with some studies reporting targeted inhibition and suppression of virulence as opposed to indiscriminate toxicity [190]. This presents an exciting new frontier for future research and therapeutic strategies for these prevalent and challenging diseases.

### 5.1. Bismuth (Bi)

Bi compounds have been utilised for many years to treat gastrointestinal disorders, including *H. pylori* infections, which are commonly associated with gastritis, and peptic ulcer disease, and are a well-established risk factor associated with the development of gastric mucosa-associated lymphoid tissue (MALT) lymphoma and gastric cancer [191,192,193]. In fact, due to the acidic environment of the stomach, it was also historically thought to be a sterile organ until the landmark discovery of *H. pylori* infection and its association with gastric disease [194]. *H. pylori* is a Gram-negative, microaerophilic bacteria which colonises the gastric mucosa of over half the world’s population, making it one of the most widespread bacterial infections [195]. Although 80% of *H. pylori*-infected individuals remain asymptomatic, some develop chronic gastritis, peptic ulcers, and eventually gastric cancer. Gastric cancer is the fourth leading cause of cancer-related deaths globally, and *H. pylori* infection accounts for nearly 90% of all non-cardia gastric adenocarcinomas [196]. Eradication of *H. pylori* infection can reduce the risk of gastric cancer development, especially if treated early before the onset of precancerous lesions [197]. The standard treatment for *H. pylori* infection is a combination of proton pump inhibitors (PPIs) and antibiotics (clarithromycin, amoxicillin, or metronidazole). The emergence of antibiotic resistance has led to a decline in the effectiveness of these regimens [198]. However, the synergistic effect between Bi salts and antibiotics has been observed, making Bi-containing quadruple therapy a recommended first-line treatment in areas with a high prevalence of antibiotic resistance [199]. Bi has a multi-targeted mode of antimicrobial activity by disrupting the bacterial cell envelope, interfering with enzyme function, inhibiting bacterial protein synthesis, and disrupting nickel homeostasis [40,200]. Nickel is essential for the survival and pathogenesis of *H. pylori,* as it regulates nickel acquisition, storage, delivery, and efflux via the synthesis of various metalloproteins/chaperones [201]. Bi drugs can interfere with nickel homeostasis by binding to nickel-associated proteins that play a critical role in urease and [Ni,Fe]-hydrogenase maturation, leading to the inhibition of enzyme activity. It has been widely reported that in addition to antimicrobial activity, Bi compounds exhibit anti-inflammatory and gastroprotective properties, which contribute to their effectiveness in treating these conditions and highlight their potential as a microbiome modulator [202,203].

### 5.2. Gold (Au)

Auranofin, an Au-based compound initially developed and approved for rheumatoid arthritis, has recently been recognised as a promising microbiome modulator. The antimicrobial potency of auranofin has been demonstrated against a broad spectrum of pathogenic bacteria, including antibiotic-resistant strains, such as *Clostridioides difficile*, *M. tuberculosis*, methicillin-resistant *S. aureus* (MRSA), and vancomycin-resistant *Enterococcus* (VRE), both in vitro and in vivo [49,204,205,206]. The modus operandi of its antimicrobial action is its ability to disrupt the redox balance within bacteria by inhibiting the TrxR enzyme, an essential component of their antioxidant defence mechanism [205]. Additionally, auranofin has been shown to have anti-virulence properties, reducing the production of key virulence factors, including proteases, lipase, and haemagglutinin [207]. Interestingly, auranofin’s influence on the microbiome extends beyond its antimicrobial properties. The inherent anti-inflammatory properties of the drug also hold implications for microbiome modulation. It achieves this by curtailing the expression of pro-inflammatory cytokine IL-6 via the inhibition of the NF-κB-IL-6-STAT3 signalling cascade, a critical pathway involved in the pathogenesis of various inflammatory diseases [208]. By mitigating local inflammation, auranofin maintains the integrity of the host barrier, thereby fostering a more conducive environment for beneficial bacteria. For instance, in murine models, auranofin treatment led to decreased inflammation and a shift towards a more balanced microbiota, characterised by an increase in anti-inflammatory bacterial strains such as *Faecalibacterium prausnitzii* [209,210,211]. Thus, auranofin presents a dual action microbiome modulator, harnessing both antimicrobial and anti-inflammatory capabilities. This novel mechanism of action offers a promising avenue for therapeutic intervention in conditions characterised by microbial imbalance, although further research is warranted to fully elucidate its potential clinical utility. Moreover, recent studies have focused on auranofins’ potential asan anticancer agent, showing its efficacy against various cancers, including lung cancer (IC50 < 2 μM, for A549) [169,212,213,214], while also enhancing Ibrutinib (tyrosine kinase inhibitor) activity in EGFR-mutant lung adenocarcinoma [215]. Mimicking its antimicrobial mechanism of action, auranofin is a selective inhibitor of TrxR, triggering increased production of ROS and activating the p38 mitotic activated protein kinase (p38 MAPK) [216]. It has also been reported that auranofin can inhibit proteasome-associated deubiquitinases (DUB), deregulating the ubiquitin-proteasome system (UPS) [217]. The proven effectiveness of auranofin in cancer management has sparked interest among pharmaceutical chemists in exploring other Au(I) complexes for their potential therapeutic roles. The future of Au-based drugs, thus, presents an intriguing avenue for cancer treatment, in particular cancer that arises from microbiome dysbiosis. 

### 5.3. Silver (Ag)

Silver has long been known for its potent antimicrobial properties, both historically and in modern times. A variety of medical products containing silver are available, such as bandages, ointments, and catheters in the form of nanocrystalline silver (including silver nanoparticles and colloidal silver), silver nitrate, and silver sulfadiazine (a complex formed with the antibiotic sulfadiazine) [218]. Ag(I) compounds have well-documented multi-modal properties that exhibit broad-spectrum activity against a wide range of bacterial species, including Gram-positive and Gram-negative bacteria, as well as fungi and viruses [219,220]. The toxicity of Ag(I) compounds primarily stems from the release of ions that interact with the cell envelope and destabilise the membrane [221,222], coupled with nucleic acids and proteins disrupting replication and synthesis [223] and inhibiting metabolic pathways [199,224]. Although Ag(I) is generally not regarded as redox-active, the generation of ROS is also attributed to its antibacterial activity [35,48,219]. However, it is thought that the production of ROS indirectly occurs through the perturbation of the respiratory electron transfer chain [225], Fenton chemistry following destabilisation of Fe-S clusters or displacement of Fe [226] and inhibition of anti-ROS defences by thiol–Ag bond formation [227]. Studies have also shown that Ag(I) often exhibits synergistic effects when combined with a range of antibiotics, such as β-lactams [228,229,230], aminoglycosides [48,230,231,232] and fluoroquinolones [229,230], and tetracyclines [48], against both planktonic and biofilm forms. While the direct antimicrobial effects of Ag(I) are well documented, its role as a microbiome modulator has been an area of growing interest in recent years and has produced contradictory results thus far. An in vivo study evaluating changes in the populations of intestinal-microbiota and intestinal-mucosal gene expression in rats after oral administration of Ag nanoparticles (AgNP) (9, 18, and 36 mg/kg body weight/day) and silver acetate (100, 200, and 400 mg/kg body weight/day). The results indicate that exposure to AgNP prompted size- and dose-dependent changes to ileal mucosal microbial populations, as well as intestinal gene expression, and induced an apparent shift in the gut microbiota toward greater proportions of Gram-negative bacteria [233]. In contrast, another study found non-significant alterations in the Bacillota (Firmicutes) and Bacteroidota populations with no toxicological effects on rats that received AgNPs orally by gavage for 28 days [234]. Various studies have reported that the shape of AgNPs can influence their impact on gut microbiota. For example, cubic AgNPs have been shown to reduce the abundance of Christensenellaceae, *Clostridium* spp., *Bacteroides uniformis*, and *Coprococcus eutectic* [235], while spherical AgNPs decrease the presence of *Oscillospira* spp., *Dehalobacterium* spp., *Peptococcaceae*, *Corynebacterium* spp., and *Aggregatibacter pneumotropica* populations [236]. Ag(I) complexes have also been studied for their potential use as anticancer agents due to their unique properties and interaction with cellular components, such as DNA and proteins, leading to disruption of essential biological processes [237]. For example, Ag(I) complexes with N-heterocyclic carbene ligands have been studied for their anticancer activities against a range of cancer cell lines, including lung cancer [238].

## 6. 1,10-Phenanthroline and Its Metal Complexes

The coordination of a metal ion to a biologically active ligand can serve to facilitate the uptake of the non-lipophilic metal across the lipophilic cell envelope or to act synergistically with the metal, such that the combined toxic effects of the metal and the active ligand will exert enhanced and targeted activity in the problematic cell [35,189]. 1,10-Phenanthroline (phen) is a heterocyclic and chelating bidentate ligand for metal ions, which has played an important role in advancing coordination chemistry. The ideally placed nitrogen atoms, as seen in Figure 1, have rigid planar structure, hydrophobic, π-electron-deficient heteroaromatic, and acidic properties, allowing the ligand to assist in the stabilisation of a variety of metal complexes in various oxidation states [239,240]. Phen has long shown antibacterial effects in an in vitro setting, demonstrating excellent bacteriostatic activity on Gram-positive and Gram-negative species of pathogenic bacteria [241,242]. This antimicrobial action has been attributed to the chelating capabilities of phen and the sequestering nature of metal ions [243]. Thus, it selectively disturbs the essential metal cellular metabolism interference with metal acquisition and bioavailability for crucial reactions impeding the microbial nutrition, growth, virulence, and pathogenesis of a variety of microorganisms, including *Leishmania* spp., *Aspergillus* spp., *Candida albicans*, *Fonsecaea pedrosoi,* and *Streptococcus agalactiae* [244,245]. Accordingly, metal chelators, such as phen, have been investigated for their therapeutic potential against microbial infections, as metals such as Fe, Cu, and zinc (Zn) play an important role in cellular homeostasis [190]. Moreover, it was shown that the sequestered metals produce a metal-phen complex, and the emerging complex drives the toxic effects [246]. Investigations into the in vitro antibacterial activity of phen derivatives (Figure 1), such as 3,4,7,8-tetramethyl-1,10-phenanthroline, 5-nitro-1,10-phenanthroline, 1,10-phenanthroline-5,6-dione, 2,9-dimethyl-1,10-phenanthroline, and various others have also been undertaken [247]. A significant increase in biocidal activity was achieved when the various ligands were coordinated with Cu(II) ions, with the 2,9-dimethyl derivative being the most active against *S. aureus* and *Escherichia coli.*

Many anticancer metal complexes with cytotoxicity have been reported based on phen bidentate ligands [248,249,250,251,252]. Compounds with the general formula [Cu(L-dipeptide)(phen)]·nH_2_O have been screened for anticancer activity in lung (A549) cancer cells showing anticancer potencies in the micromolar concentration range [253]. Mixed Cu(II) phen-based complexes with the general formula Cu(N-N^1^)_x_(OH_2_)_y_(ClO_4_)_z_, where N-N^1^ = phen, were reported to be toxic towards the squamous cell carcinoma (SKMES-1) cell line ranging from picomolar to micromolar in terms of their IC_50_ [254,255]. The mechanism by which Cu(II)-phen complexes exert their toxicity is reported to be through strong DNA binding activity and by inducing oxidative stress through mitochondrial dysfunction initiating apoptosis [249,250]. A series of Au(III) bearing phen and derivative 2,2′-bipyridine (bipy) scaffolds have shown promising anticancer activity against the A549 cell line, which exhibited IC_50_ values that were significantly lower than those of cisplatin control, possibly inducing cell death via a TrxR-mediated mechanism [256]. The coordination complexes [Au^III^(5-chloro-phen)Cl_2_]PF_6_) and [Au^III^(bipy)Cl_2_]PF_6_ displayed potent cytotoxicity in the A549 cell line, potentially through the inhibition of the water and glycerol channel aquaglyceroporin-3 (AQP3), which play crucial roles in cell apoptosis, proliferation, and migration and therefore have been proposed as new drug targets for cancer treatment [257]. Ru-phen complexes, such as [Ru(phen)_2_dppz]^2+^ (where dppz=dipyrido[3,2-a:2′,3′-*c*]phenazine), has been shown to intercalate with DNA and induce photocleavage, leading to significant cytotoxicity in various cancer cell lines, including lung cancer [258]. The octahedral complex [Ru(phen)_3_]^2+^ can intercalate and unwind DNA as effectively as ethidium bromide [259]. Silver(I) complex Ag(Phen)_2_(CH_3_COO)· H_2_O exhibits anticancer properties against lung adenocarcinoma (A549) cells through ROS production, which in turn induces a change in the mitochondrial membrane potential [260]. Overall, the exact mechanism of action of these metal-phen complexes in the context of lung cancer still requires further exploration. More comprehensive in vitro and in vivo studies, followed by clinical trials, are necessary to fully understand their potential.

Metal complexes containing phen, and its derivatives, have also been reported in the literature for their antifungal [244,261,262,263,264], antiparasitic [265,266,267,268], antiviral [269,270,271,272], and antibacterial [273,274,275,276,277] competence. The work of polypyridyl metal complexes was pioneered by Francis Dwyer and co-workers when they published a landmark study outlining the biological activity of dicationic Ru, Fe, nickel (Ni), cobalt (Co), Cu, Zn, calcium (Ca), and manganese (Mn) chelates containing phen and its derivatives [278]. Their work established the in vitro toxicity of the metal-phen chelates against *M. tuberculosis*, *S. aureus*, *S. pneumonia*, *Clostridium perfringens*, *E. coli*, and *Proteus vulgaris* (presented in descending order of activity), while the metal-free phen demonstrated dampened effects. Furthermore, the toxicity exerted was independent of the metal present, except for *M. tuberculosis*. They also identified that the bacteria and fungi did not develop any significant resistance to the selected chelates. In vivo bacterial infection treatment studies using mice and guinea pig models showed that metal-phen chelates were clinically useful as topical antimicrobials. However, the selected compounds were chemotherapeutically ineffective when administered intravenously due to rapid clearance from the bloodstream [278]. Although the results were promising, interest in polypyridyl metal complexes as potential antimicrobial chemotherapeutics was void in the pharmaceutical sector, possibly due to the vast array of antimicrobials in the pipeline or the immensely lower incidence of resistance at the time. In modern times, transition metal-based compounds have had a revival of interest as possible alternatives or adjuvants to the current armamentarium of antimicrobial agents that cannot contend with the rapid emergence of resistant microbes worldwide [35].

## 7. Mechanisms of Metal-Phen Complexes

Research into the possible mechanisms by which promising phen-based complexes exert their toxic effects has been carried out within in vitro models, including mammalian, fungal, and bacterial cells, and in a range of in vivo biological models. McCann et al. [279] proposed modes of antifungal action by phen and its derivatives to be (i) the dysfunction and disruption of the cell membrane along with the withdrawal of the cytoplasmic membrane, (ii) drug-induced circumvention on the action of cell division (budding), (iii) damage to the functional mechanisms of the mitochondria, (iv) chelation or sequestering of trace metal ions which inhibits glycosylphosphatidylinositol (GPI) biosynthesis, (v) rupturing of internal organelles along with the enlargement of the nucleus, and (vi) the degrading of nuclear DNA [279]. The following sections discuss the potential antibacterial capability of metal-phen complexes and their mechanism of action.

### 7.1. The Bacterial Cell Envelope and Activity of Metal-Phen Complexes

The cell wall and outer cell membrane are considered a significant obstacle in developing novel antibacterial agents that are effective against Gram-negative bacteria, as they must be lipophilic and able to penetrate the outer membrane. Contradictory to this, Ag(I) has well-documented bactericidal properties. In its cationic form, Ag(I), it is oligodynamic and displays a broad spectrum of activity that is dependent upon the slow release of biologically active ions thought to bind to the bacterial cell surface and interfere with growth by inhibiting transport across the cell wall and membrane [35,219]. Being charged entities, free metal ions require protein- or ionophore-mediated transport to cross a lipid bilayer, as these transporter proteins encase the cations in a hydrophobic sleeve to enable its passing. The complexation of metals to a hydrophobic chelating ligand such as phen can enact the same process, enabling their penetration through the bacterial cell envelope, presenting the chelate to its target biomolecule to inhibit cell growth or initiate cell death [280]. Cationic metal-phen chelates can be bacteriostatic [278] or bactericidal [281] towards many Gram-positive bacteria, including *S. aureus* and *S. pyogenes* and *C. perfringens*. However, they do not exhibit the same potency against Gram-negative bacteria. The lipophilicity of a complex is taken as a good measure of its ability to pass into the cell by diffusion, and in some cases, increased lipophilicity correlates with antimicrobial potency [282]. Dwyer et al. [283] were the first to identify this relationship when working with mononuclear Ru(II) and the phen complex, [Ru(phen)_3_]^2+^. Originally, this complex was shown to be inactive against all tested bacterial strains; however, when methyl substituents were incorporated into the phen ligand, [Ru(3,4,7,8-Me_4_phen)_3_]^2+^, the non-polar surface interaction was increased. This corresponded to an increase in activity against all examined bacteria, especially Gram-positive and acid-fast bacteria [283]. There has been renewed interest in the antimicrobial activity of polypyridylruthenium(II) complexes over the past decade. Crowley et al. [284] have used a series of 2-(1-R-1*H*-1,2,3-triazol-4-yl)pyridine “click” ligands (R-pytri) as functionalised analogues of 2,2′-bipyridine (bpy) and phen chelators to synthesise a diverse range of ruthenium complexes ([Ru(R-pytri)_3_](PF_6_)_2_), and examined their antimicrobial activities. Some complexes demonstrated moderate activity against Gram-positive strains, but this was not maintained when examined against Gram-negative bacteria. Ru(II) complexes [Ru(L)_2_amtp]^2+^ (L = bpy) [285] and [Ru(bpy)_2_L] (L = *p*-tFMPIP) [286] were also active against Gram-positive bacteria, particularly *S. pneumoniae* which is a problematic pathogen for COPD patients. The mode of action was reported to be interference with iron acquisition systems in *S. pneumoniae* cells [285], oxidative stress and membrane damage [286]. Moreover, the Ru(II) complexes were not toxic towards human bronchial epithelial cells [286] or A549 cell line [285]. Keene, Collins, and co-workers have extensively researched Ru(II) complexes and their potential as antimicrobial agents. They developed kinetically inert mono-, di-, tri-, and tetra-nuclear polypyridylruthenium(II) complexes, whereby the Ru(II) metal centres are bridged by flexible and lipophilic bis [4(4′-methyl-2,2′-bipyridyl)]-1),*n*-alkane ligand (bb_n_) and are collectively termed ‘Rubb_n_’ complexes, where n = the number of methyl groups in the bb_n_ linker chain (n = 2, 5, 7, 10, 12, 14 or 16). The antibacterial activity of this series of complexes is correlated with increasing lipophilicity through increased length of the bb_n_ chain and only slight differences were observed with enantiomeric forms of the complexes [285]. The dinuclear ruthenium complexes ‘Rubb_n_’ have been the most extensively studied and have produced exciting results. The Rubb_n_ complexes enter bacteria in an energy-dependent manner as they significantly depolarise and permeabilise the cellular membrane [286]. There was preferential uptake of Rubb_n_ in prokaryotes compared to cancerous cells, which was suggested to result from the greater presence of negatively charged components in the bacterial envelope [287]. Rubb_16_ was found to be the most active with selective toxicity towards bacteria. This complex condensed ribosomes when they existed as polysomes by preferentially binding to RNA over DNA in vivo, offering the interruption of protein synthesis in actively growing bacterial cells as a potential mode of action [288].

The corresponding tri- and tetra-nuclear complexes ‘Rubb_n_-tri’ and ‘Rubb_n_-tetra’ were the more active, demonstrating MIC’s four times their dinuclear analogues [289]. Although the level of cellular uptake in Gram-negative bacteria was similar to that of Gram-positive bacteria, there was significantly less activity in the former species. The authors suggested that this was a result of the inherent resistance of Gram-negative bacteria to inert polpyridylruthenium(II) complexes, particularly *P. aeruginosa*, in which the outer membrane permeability is 10- to 100-fold lower than, for example, that of *E. coli* [290]. However, while the antibacterial activity increased as the ruthenium centres and the length of the alkyl chain in the bb_n_ ligand increased, the toxicity towards eukaryotic cells reduced selectivity [291].

### 7.2. DNA as an Antibacterial Target for Metal-Phen Complexes

The design of agents capable of controlled nucleic acid cleavage is of great importance, and since the initial work of Sigman et al. [292], there has been considerable attraction to artificial metallonucleases. The copper-bis-phenanthroline complex, [Cu(phen)_2_]^2+^, induced oxidative cleavage of DNA in the presence of a reductant, which is unusual for complexes containing phen, as this compound as a singular agent usually intercalates with DNA. DNA as a target offers a fresh avenue for potential antibacterial agents, as it has been relatively unexplored thus far [293]. A large number of publications have reported the enhanced antibacterial activity of quinolone/fluoroquinolone antibiotics containing metal(II)-phen complexes (metal = Cu(II), Ni(II), Co(II), Mn(II)) [294]. One example is the combination of Levofloxacin (lvx) with Cu(II), forming the binary complex, [Cu(lvx)]^2+^ that significantly increases DNA binding but is not stable at a physiological pH. However, the addition of phen as an *N*-donor forms a very stable ternary complex [Cu(lvx)(phen)]^2+^ [295]. Cu(II)/phenanthroline/fluoroquinolone complexes have demonstrated intercalation with DNA, exhibiting nuclease activity, and they are taken into the cell differently from that of the free fluoroquinolone drug [296]. Furthermore, when tested against a mutant strain of *E. coli* lacking porins, it was identified that the higher the hydrophobicity of the complexes, the higher the need for porins for their uptake [297]. Marmion et al. [298] rationally developed a family of metallo-antibiotics with the general formula [Cu(N,N)(CipA)Cl], where N,N represents a phenanthrene ligand and CipA is a derivative of fluoroquinolone antibiotic ciprofloxacin. The complexes were active against susceptible and resistant *S. aureus* bacteria, which were identified in the lungs of COPD patients during exacerbation. They appear to intercalate DNA via minor groove interactions and mediate DNA damage by generating ROS with superoxide and hydroxyl free radicals playing crucial roles in DNA strand scission [298]. Molecular docking analysis prompted the synthesis of derivatives [Cu(N,N)(CipHA)]NO_3_, where CipHA represents a hydroxamic acid of ciprofloxacin. Proteomic analysis of *S. aureus* exposed to two lead complexes [Cu(phen)(CipHA)]NO_3_ and [Cu(DPPZ)(CipHA)]NO_3_ (where DPPZ = dipyridophenazine) suggests that proteins involved in virulence, pathogenesis, and the synthesis of nucleotides and DNA repair mechanisms are most affected [299]. Metal-phen complexes without the addition of the quinolone ligand have also demonstrated moderate antibacterial activity, with DNA binding or nuclease activity as the proposed antibacterial mechanism of action. The metals include Ag(I) [300], Cu(I) [301], Cu(II) [302], Zn(II) [303], Pt(II) [304], Mn(II) [305], and Fe(III) [306].

### 7.3. The Activity of Metal-Phen Complexes on Biofilms

Bacterial biofilm communities differ from planktonic bacteria in various ways, such as growth rate, gene expression, and protein expression [307]. This is due to biofilms creating an altered microenvironment with higher osmolarity, nutrient scarcity, and increased cell density of heterogeneous bacterial communities [308,309]. Bacteria usually reside in biofilms, and biofilm-residing bacteria can be resilient to the immune system, antibiotics, and other treatments [307,310,311]. Such biofilms enable bacteria to persist in the lower respiratory tract, which can exacerbate the disease and complicate the treatment of patients with COPD and lung cancer [312,313,314]. Therefore, agents that can navigate the difficult terrain of a biofilm to the bacteria embedded within or dissociate the extracellular matrix to expose the bacteria are important.

Although there are few reports of metal-phen complexes with anti-biofilm activity, there have been some advances in the development of novel compounds. A range of Cu(II) complexes, [Cu(I_L_)(A_L_)]^2+^ (where I_L_ represents functionalised phens and A_L_ represents 1S,2S-1,2-diaminoethane or 1R,2R-diaminocyclohexane), were tested on a number of bacterial strains [315]. Although the metal complexes generated higher MICs (2–32 µg/mL) than the control antibiotic vancomycin (MIC = 0.25 µg/mL), they showed significant activity against *S. aureus* biofilms and were better at removing biofilms than vancomycin. For example, 100 µg/mL of vancomycin, which is 400-fold the MIC, reduced biofilm biomass by just 44%, whereas 25 µg/mL of [Cu(5,6-dimethyl-phen)(*SS*-dach)]^2+^ (equivalent to 3-fold the MIC) reduced the biofilm by 68% in only 2 h. This decrease in biomass was similar in all Cu complexes and is metal-dependent, as replacing the centre with Pt(II) or Pd(II) removed both the antibacterial and anti-biofilm action. Therefore, this suggests that the potential mechanism of action in both planktonic and biofilm cells is related to the nuclease activity of Cu, as neither Pt nor Pd possesses this capability, particularly given that the extracellular matrix contains a considerable quantity of nucleic acids [315]. Similarly, Mn(II), Cu(II), and Ag(I) complexes incorporating phen and 3,6,9-trioxaundecanedioate (tddaH_2_) (Figure 2) showed enhanced activity profiles when tested against clinical isolates of *P. aeruginosa* [232], a bacteria frequently reported to be problematic to both COPD and lung cancer patients. The results showed that the metal-tdda-phen complexes could prevent biofilm formation, in relation to mass and cellular viability, to a greater capacity than gentamicin across the clinical strains and disturb mature biofilm. This was supported by reducing the separate biofilm components examined, suggesting extracellular DNA (eDNA) and extracellular polysaccharides as potential molecular targets [232]. The ability to act on *P. aeruginosa* clinical isolates synergistically with gentamicin on mature biofilms prompted in vivo studies using *G. mellonella* larvae [316]. Mn-tdda-phen and Ag-tdda-phen were able to clear a *P. aeruginosa* infection at concentrations that are non-toxic towards *G. mellonella* larvae in a multi-modal fashion by acting directly on the bacteria in addition to stimulating both the cellular (hemocytes) and humoral (immune-related peptides, specifically transferrin and inducible metalloproteinase inhibitor) immune response of the larvae. The amalgamation of metal-tdda-phen complexes and gentamicin further intensified this response at lower concentrations, clearing a *P. aeruginosa* infection that was previously resistant to gentamicin alone [316]. The same complexes have also been reported to have antitubercular activity and were highly toxic towards the NSCLC (A549) cell line [274]. Gandra et al. [261] reported on the antifungal capabilities of metal-phen complexes against isolates of three species that make up the highly resistant *Candida haemulonii* species complex, an emerging opportunistic pathogen that has been reported to be problematic in COPD and lung cancer [317,318]. Mn(II)- and Ag(I)-phen chelates could conserve antifungal activity at concentrations that were reasonably non-toxic toward *G. mellonella*. Most notable was the Mn-tdda-phen complex, as it induced the lowest mortality rate while reducing the fungal burden on infected larvae and could also affect the virulence of *C. haemulonii* [262]. Across all studies, the inclusion of phen was paramount to the potency of the complexes, with the addition of tddaH_2_ enhancing their water solubility and mode of action in various microbial cells, which are problematic in chronic lung diseases, particularly highlighting COPD and lung cancer. A series of complexes incorporating phen and cyanoguanidine (cnge) have been reported with the general formula M(II)/phen/cnge (where M = Cd, Cu, or Zn) [319]. The cadmium complex [Cd(phen)_2_(SO_4_)H_2_O]cnge· 5H_2_O possesses enhanced activity across the assessed bacterial and fungal pathogen in comparison to its metal derivatives. This prompted subsequent anti-biofilm analysis against *P. aeruginosa* laboratory strain ATCC 27853, in which 0.5 x MIC (93.5 µg/mL) of the metal complex inhibited approximately 40% of biofilm formation. Moreover, there was a reduction in the acute toxicity of the phen ligand when it was incorporated into the Cd(II)/phen/cnge complex within a crustacean model, *Artemia salina*.

A derivative of phen, 1,10-phenanthroline-5,6-dione (phendione, Figure 1), has also been investigated for its antibacterial and anti-biofilm capabilities. This compound contains an *o*-quinoid moiety and forms strong complexes with transition metal ions at the *N-N* terminal, with predominance toward Zn(II) and to a lesser extent for Fe(II), Ca(II), Cu(II), Co(II), and Mn(II) [279]. Tay et al. [320] reported the MIC and minimum bactericidal concentration (MBC) values of phendione for *Enterococcus faecalis* as 2 μg/mL and 16 μg/mL, respectively, relating its activity to its ability to sequester Zn(II) from metalloenzymes. In order to kill *E. faecalis* cells embedded in a biofilm structure, an MIC four times that required to kill planktonic bacteria was required. However, the biofilm was eradicated at this concentration. Although the authors could not explain the mechanism by which phendione eradicates *E. faecalis* biofilm, they speculated that it may weaken the extracellular polymeric substances of the biofilm by disrupting Zn(II) balance. The metal-free phendione and coordinated Cu(II) and Ag(I) complexes, [Ag(phendione)_2_]ClO_4_ (Ag-phendione) and [Cu(phendione)_3_](ClO_4_)4H_2_O (Cu-phendione) (Figure 3) have been extensively studied across a variety of microbial cells [321]. The metal-phendione complexes were able to inhibit the growth of the *Phialophora verrucosa* [263,264], *Pseudallescheria boydii* [244], *Trichomonas vaginalis* [266] and *Candida albicans* [279]. Viganor et al. [273] investigated the effect of metal-phendione complexes on planktonic and biofilm-growing *P. aeruginosa*. The compounds presented the following potency against susceptible and resistant planktonic cells: Cu-phendione (7.76 µM) > Ag-phendione (14.05 µM) > phendione (31.15 µM) > phen (579.28 µM). It was also discovered that the compounds could disrupt a mature biofilm in a dose-dependent manner, especially Ag-phendione (IC_50_ = 9.39 µM) and Cu-phendione (IC_50_ = 10.16 µM). The metal-phendione complexes were reported to cause a significant reduction in the expression of the metalloenzyme Elastase B produced by *P. aeruginosa* at a gene and mature protein production level [322], therefore suggesting this as a potential therapeutic target. Furthermore, the complexes also offer protective action for lung epithelial cells. Metal-phendione complexes can interact with double-stranded DNA and promote oxidative damage, suggesting multiple mechanisms of action in *P. aeruginosa* [323]. The same activity sequence of the test complexes (Cu-phendione > Ag-phendione > phendione) was maintained when assessed in both planktonic- and biofilm-forming cells of MDR *Acinetobacter baumannii* [276] and *Klebsiella pneumoniae* [277] clinical isolates. The combination of either Cu-phendione or Ag-phendione with meropenem had synergistic activities, according to their fractional inhibitory concentration, against *Klebsiella pneumoniae* carbapenemase (KPC)-producing *K. pneumoniae* clinical isolates. Moreover, the combination of metal complex and meropenem restored the antibiotic’s efficacy (in terms of MIC) in 87% of phenotypically resistant strains [277]. The metal-phendione complexes also had low toxicity in *G. mellonella* larvae [262,264] and mice [243], reinforcing that these compounds may represent a novel antimicrobial agent with potential therapeutic applications across a variety of pathogens that predominate in chronic lung disorders.

## 8. Conclusions and Future Perspectives

The field of lung microbiome research is rapidly expanding, but faces numerous challenges [7,324,325,326]. The low microbial biomass and high host contamination make it challenging to use metagenomics and metatranscriptomics to understand the haemostatic and pro-oncogenic functions of the microbiome. Secondly, chronic lung diseases are heterogeneous, making it necessary to untangle the complex relationships between microbiome, disease phenotypes, and endotypes. This understanding is crucial for determining the role of the microbiome in the development and progression of these diseases, including the transition to cancer. Thirdly, unlike the well-characterised gut microbiome, little is known about the specific mediators produced by the lung microbiome and their functions. A systems biology approach is required to study the interaction between airway microbes and host in diseased states. There needs to be a standard procedure for manipulating airway microbiota in animal studies, which is crucial for assessing its functional impacts. While NGS is powerful, culturing microbes from the respiratory tract is essential for translational research to gain a comprehensive understanding of the pro-oncogenic pathways that are being activated. Overall, these challenges hinder our understanding of the lung microbiome and its potential implications for health and disease progression.

Metal-based drugs, including metal-phen complexes, have emerged as promising candidates for managing chronic lung diseases and related microbial complications [38,200,327,328]. These drugs have demonstrated significant antimicrobial, anti-biofilm, and anti-virulence activity, targeting multiple pathways in pathogenic bacteria, distinct from traditional antibiotics. The potential of these complexes to act against antibiotic-resistant strains and their synergy with existing antibiotics could also help tackle the pressing issue of antibiotic resistance. While these avenues are promising, it is essential to note that the development of metal-based drugs as therapies for microbiome modulation is in its nascent stages. Moreover, the link between microbiome modulation and cancer prevention remains a complex issue that requires more research [118,120,329,330]. While it is known that chronic inflammation can contribute to cancer development and that bacterial imbalances in the lungs can drive this inflammation, it is not yet entirely clear how effectively and consistently altering the microbiome can reduce cancer risk. Nevertheless, the versatility of metal drugs and metal-phen complexes and their potential for further chemical modification opens new avenues for developing effective antimicrobial and anticancer agents. While it is promising that metal complexes are currently under evaluation in clinical trials for other indications, it is crucial to substantiate whether these metal-based drugs can effectively safeguard humans against pathogenic bacteria. Furthermore, it is critical to consider that while metal-based drugs may successfully kill or inhibit the growth of pathogenic bacteria, they may also impact beneficial bacteria within the lung microbiome, potentially leading to unintended consequences.

The future of research in the lung microbiome requires innovative experimental and analytical strategies to overcome these challenges. Longitudinal, interventional, and mechanistic studies are needed to determine causality. These studies aim to address fundamental scientific questions about the lung microbiome, such as its baseline status in healthy individuals, how it responds to environmental factors, its roles in different types of respiratory diseases, and how its dysregulation can advance disease to cancer. Researchers also hope to determine whether the microbiome can be used as a marker for the diagnosis, phenotyping, and prognosis of respiratory diseases. Additionally, they seek to understand the topographic structure and spatial dynamics of microbial communities in the lung and the interactions and influences between respiratory bacteria, fungi, and viruses on the host immune system. Identifying key microbial metabolites that regulate host inflammation or other processes in the respiratory tract is of interest. Ultimately, the goal is to use the airway microbiome as a biomarker and therapeutic target for precision medicine in respiratory and broader human diseases.

To conclude, the potential of metal-based drugs to serve as novel therapeutics for COPD and lung cancer is compelling. By deepening our understanding of their interaction with the diverse bacterial communities within these conditions and through rigorous clinical testing, we can pave the way for innovative treatments that could significantly improve patient outcomes.

## Figures and Tables

**Figure 1 ijms-24-12296-f001:**
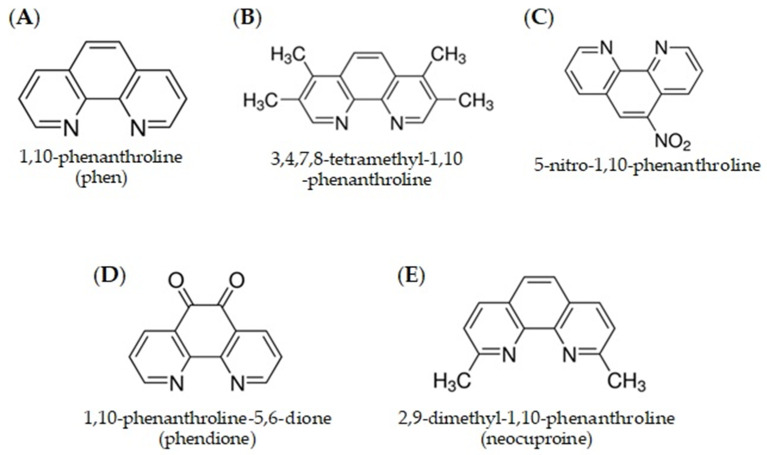
Structure of (**A**) 1,10-phenanthroline (phen), and examples of its derivatives, (**B**) 3,4,7,8-tetramethyl-1,10-phenanthroline, (**C**) 5-nitro-1,10-phenanthroline, (**D**) 1,10-phenanthroline-5,6-dione (phendione) and (**E**) 2,9-dimethyl-1,10-phenathroline (neocuproine).

**Figure 2 ijms-24-12296-f002:**
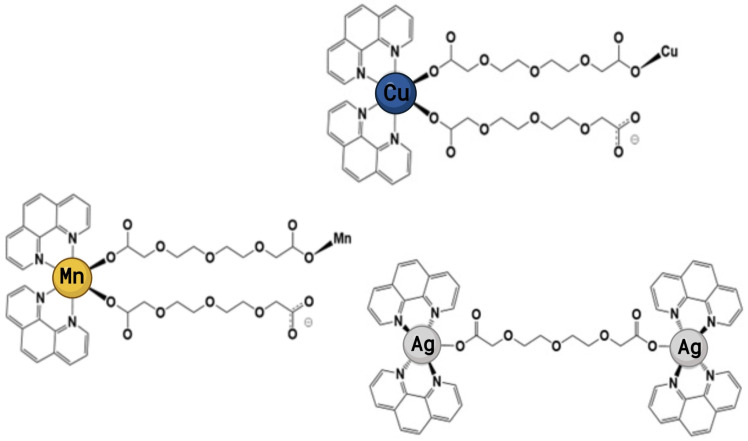
Metal-tdda-phen complexes: {[Cu(3,6,9-tdda)(phen)_2_]·3H_2_O·EtOH}_n_ (Cu-tdda-phen), {[Mn(3,6,9-tdda)(phen)_2_]·3H_2_O·EtOH}_n_ (Mn-tdda-phen) and [Ag_2_(3,6,9-tdda)(phen)_4_]·EtOH (Ag-tdda-phen). Adapted from [261].

**Figure 3 ijms-24-12296-f003:**
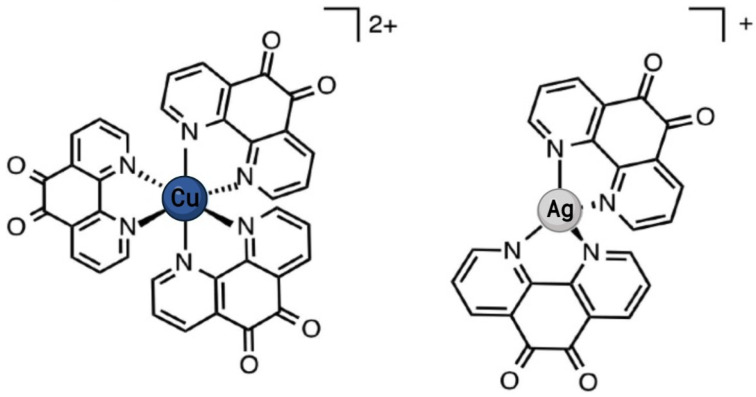
Structure of phendione containing metal complexes [Cu(phendione)_3_]^2+^ and [Ag(phendione)_2_]^+^. Adapted from [321].

**Table 1 ijms-24-12296-t001:** Food and Drug Administration (FDA) and European Medicines Agency (EMA) approved metal-containing drugs, their therapeutic applications, and year of approval.

Metal	Drug	Therapeutic Application	Year Approved
Platinum (Pt)	Cisplatin	Cancer: lung, testicular, ovarian, bladder	1978
	Carboplatin	Cancer: lung, ovarian, head and neck	1989
	Oxaliplatin	Cancer: colorectal	1999
Gold (Au)	Auranofin	Rheumatoid arthritis	1985
	Sodium aurothiomalate	Rheumatoid arthritis	1985
Iron (Fe)	Deferoxamine	Iron chelation (iron overload)	1968
	Deferiprone	Iron chelation (iron overload)	1999
	Deferasirox	Iron chelation (iron overload)	2005
	Ferric carboxymaltose	Iron deficiency anaemia	2007
	Iron isomaltoside 1000	Iron deficiency anaemia	2009
	Ferumoxytol	Iron deficiency anaemia in chronic kidney disease	2009
	Ferric citrate	Hyperphosphatemia in chronic kidney disease	2014
	Sucroferric oxyhydroxide	Hyperphosphatemia in chronic kidney disease	2013
Bismuth (Bi)	Bismuth subsalicylate	Gastrointestinal issues (ulcers, diarrhoea, acid reflux)	1939
	Bismuth subcitrate potassium	*Helicobacter pylori* infection, peptic ulcers	1998
Lithium (Li)	Lithium carbonate	Bipolar disorder	1970
Gallium (Ga)	Gallium nitrate	Hypercalcemia of malignancy	2001
Lanthanum (La)	Lanthanum carbonate	Renal failure	2004

## Data Availability

Not applicable—no new experimental data were created.

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
