# Peer review of "The Lung Microbiome in COPD and Lung Cancer: Exploring the Potential of Metal-Based Drugs"

_ijms, 2023, doi:10.3390/ijms241512296_

Round 1

Reviewer 1 Report

I consider the subject of the manuscript entitled “The lung microbiome in COPD and lung cancer: exploring the potential of metal-based drugs” to be of very high interest and appropriate for publication in the International Journal of Molecular Sciences, and I recommend publication with a few minor revisions.

The paper gives an overview of the current knowledge regarding the role of the lung microbiome in both healthy individuals and its intricate implications in COPD and lung cancer, as well as the progress made in the coordination chemistry field in order to obtain metallodrugs that could improve patient outcome for those suffering from lung disease, especially metal complexes with antibacterial activity.

The article is very well organized, following a logical path from defining the issue of the role of the microbiome in lung disease and bringing into focus an alternative method that could help manage recurrent infections with pathogenic resistant bacterial species.

Although the manuscript is very well written and the information is extremely well documented, I would suggest a few minor changes be made before publication.

Firstly, although the authors focus on the many advantages of metal complexes as therapeutic agents (either already used in therapy or as potential drugs), I would consider useful to point out their downsides as well, for example the many limitations that they present in terms of adverse effects and systemic toxicity (especially the Pt and Au complexes).

The authors mention that several metallodrugs are currently in human clinical trials, it would be very interesting to give a few examples (Line 324).

The year mentioned in Table 1 for the approval of lithium carbonate for the treatment of bipolar disorder is 2002, but I think it was approved by the FDA at the beginning of the 70’s, please check that fact and the cited source.

Author Response

Dear Ms Zhang and reviewers,

The authors would like to thank the reviewers for their valuable time in providing insightful and useful contributions to this manuscript. The inputs received have helped to improve the manuscript and strengthen its impact. We have revised the manuscript and responded to the reviewers in accordance with their remarks.

Please find in below in our ‘Response to reviewers comments’, the detailed response to the individual reviewers and their comments regarding this manuscript. The response has been written in a comment point counter point format with the reviewers comments in normal font and the authors response in red font. All revisions to the manuscript have been made using track changes.

The authors look forward to hearing from you regarding our submission. We would be happy to respond to any further questions and comments you may have.

Sincere thanks,

Megan 

Response to reviewers comments

Firstly, although the authors focus on the many advantages of metal complexes as therapeutic agents (either already used in therapy or as potential drugs), I would consider useful to point out their downsides as well, for example the many limitations that they present in terms of adverse effects and systemic toxicity (especially the Pt and Au complexes).

This authors agree with this point and have therefore incorporated this into the review (lines 377-392)

The authors mention that several metallodrugs are currently in human clinical trials, it would be very interesting to give a few examples (Line 324).

This suggestion has also been incorporated, specifically regarding metal complexes that have been clinically tested against lung cancer (lines 348-376)

The year mentioned in Table 1 for the approval of lithium carbonate for the treatment of bipolar disorder is 2002, but I think it was approved by the FDA at the beginning of the 70’s, please check that fact and the cited source.

Thank you for noticing this error, the authors have amended in the document

Reviewer 2 Report

The review entitled "The lung microbiome in COPD and lung cancer: exploring the potential of metal-based drugs" highlights the role of metal-based drugs in the treatment of chronic obstructive pulmonary disease (COPD) and lung cancer. The review is structured in seven Sections, the overall composition being complete. I appreciate the effort of the authors to exploit such a complex field of medicine.

In Introduction section, the authors describe the importance of the lung microbiome in maintaining health, an alteration of that causing chronic obstructive pulmonary disease (COPD) and lung cancer. The metal-based drugs can be considerate an alternative therapeutic approach, even if, as authors mentioned, the contribution of the drugs is more as modulators.

In Sections 2 and 3, the authors showed the main factors affecting the lung microbiome. If bacteria strains are involved in lung diseases, I think that the role of antibiotics should be mentioned, more than that of the metals, or the combined therapies. Some additional information regarding these aspects must be included.

Section 4, Table 1- I consider that only metal-based drugs directly involved in the treatment of lung cancer must be included.

Section 5, only the metal-based drugs referring to the lung cancer treatment must be presented.

Section 6, I observed that metal complexes based on 1,10'-phenanthroline are mainly effective in bacteria cell activity more than in lung cancer treatment, so that some additional data regarding the efficiency of some drugs in lung cancer treatment must be added.

Based of these observations, I recommend the publication of this review after Minor revision.

Author Response

Dear Ms Zhang and reviewers,

The authors would like to thank the reviewers for their valuable time in providing insightful and useful contributions to this manuscript. The inputs received have helped to improve the manuscript and strengthen its impact. We have revised the manuscript and responded to the reviewers in accordance with their remarks.

Please find in below in our ‘Response to reviewers comments’, the detailed response to the individual reviewers and their comments regarding this manuscript. The response has been written in a comment point counter point format with the reviewers comments in normal font and the authors response in red font. All revisions to the manuscript have been made using track changes.

The authors look forward to hearing from you regarding our submission. We would be happy to respond to any further questions and comments you may have.

Sincere thanks,

Megan 

Response to reviewers comments

In Sections 2 and 3, the authors showed the main factors affecting the lung microbiome. If bacteria strains are involved in lung diseases, I think that the role of antibiotics should be mentioned, more than that of the metals, or the combined therapies. Some additional information regarding these aspects must be included.

To incorporate this suggestion the authors have added the lines 163-176, 264-267, 275-283.

Section 4, Table 1- I consider that only metal-based drugs directly involved in the treatment of lung cancer must be included.

While the authors appreciate this suggestion, they aim to also broaden the understanding of respiratory researchers who may be largely familiar with platinum-based complexes used for treatment of lung cancer, introducing them to other approved metal-based drugs. Furthermore, the rising trend of drug repurposing, or investigating the potential uses of drugs beyond their original medical indications, has been gaining momentum. An examples of which Auranofin, a gold-based drug originally approved for treating rheumatoid arthritis, has been investigated for its anticancer potential.

Section 5, only the metal-based drugs referring to the lung cancer treatment must be presented.

The authors have listened to this suggestion and added a paragraph (lines 348-392) discussing the metal drugs already approved for lung cancer and other metal drugs that have made it to clinical trial for treatment of lung cancer. The authors would like to keep the additional sections of metal drugs that have been investigated as microbiome modulators to introduce the reader to this as a concept. As research if the lung microbiome is still in its infancy the literature within this area is very limited

Section 6, I observed that metal complexes based on 1,10'-phenanthroline are mainly effective in bacteria cell activity more than in lung cancer treatment, so that some additional data regarding the efficiency of some drugs in lung cancer treatment must be added.

The authors have added the lines 583-607 to incorporate this point